# Global, regional, and national burden of cardiomyopathy (including alcoholic cardiomyopathy and others) from 1990 to 2021: An analysis of data from the global burden of disease study 2021 and forecast to 2040

**Haoyang Chen**[1,2], **Ruifeng Liang**[1]*, **Yanzhang Tian**[2]*

**1** Department of Environmental Health, School of Public Health, Shanxi Medical University, Taiyuan, Shanxi, China, **2** Department of Biliary and Pancreatic Surgery, Shanxi Bethune Hospital, Shanxi Academy of Medical Sciences, Tongji Shanxi Hospital, Third Hospital of Shanxi Medical University, Taiyuan, Shanxi, China

* ruifengliang@sina.com (RL); tianyanzhang@sxbqeh.com.cn (YT)

## Abstract

### Background

Cardiomyopathy is a disease that can lead to severe cardiac symptoms and has seen an increasing number of cases in recent years. This study aims to analyze the incidence, mortality, and disability-adjusted life years of cardiomyopathy (including alcoholic cardiomyopathy and other cardiomyopathy) globally, as well as in different regions and countries, from 1990 to 2021, at different gender, age, and sociodemographic index levels.

### Methods

All data relevant to the burden of disease analysis in this study were obtained from the 2021 Global Burden of Diseases, Injuries, and Risk Factors Study (GBD), encompassing alcoholic cardiomyopathy (AC) and other cardiomyopathy (OC). The overall burden of total cardiomyopathy (TC) was evaluated by integrating data about AC and OC, with 95% confidence intervals calculated based on the 95% uncertainty interval (UI) width divided by the standardized error value, determined by 1.96×2. Temporal patterns and trends in age-standardized prevalence rates (ASPR), mortality rates (ASDR), and disability-adjusted life years (ASR_DALYs) for global TC, as well as for AC and OC burden calculations, their estimated annual percentage changes (EAPCs) were calculated, assessed, and visualized. The analysis was categorized by gender, 20 age groups, 21 GBD regions, 204 countries/regions, and 5 socio-demographic index (SDI) regions. The burden of disease prediction model was subjected to Bayesian age-period-cohort (BAPC) modeling to derive predictions for

**Data availability statement:** All relevant data are within the paper and Supporting Information files.

**Funding:** Our research received funding from the Shanxi Provincial Department of Human Resources and Social Security (grant no. 20210002) and the Shanxi Provincial Department of Science and Technology (grant no. 202104041101024). The funders had no role in study design, data collection and analysis, decision to publish, or preparation of the manuscript.

**Competing interests:** The authors have declared that no competing interests exist.

the period from 2022 to 2040. All statistical analyses and mappings were conducted using the R statistical package version 4.4.3.

## Results

In 2021, the global burden of TC remains considerable, with a total of 4752361.3, including AC: 528429 and OC: 4223932.2; the ASPRs (per 100,000 persons) were 59.5, 6.2 and 53.3; the ASDR (per 100,000 persons) were 4.5, 0.7 and 3.7; the age-standardized DALYs were 129.8, 25.3 and 104.5. In the aspect of AC, a considerable disparity in age-standardized prevalence, mortality, and DALYs was observed between the regional, national, and gender levels. The predictive results indicated that from 2021 to 2040, the ASPR of TC and OC showed a general increasing trend, while that of AC showed a decreasing trend. The ASDR and ASR_DALYs of TC, OC, and AC showed a general decreasing trend.

## Conclusions

Globally, there has been an observed increase in the ASPR of TC and OC, while AC presented a decreasing trend, with significant regional, age, and gender variations in future trends. Although future projections in this study suggested a decline in ASDR and ASR_DALYs of TC, AC, and OC, there is a need to continue controlling the burden of disease in TC, AC, and OC studies to respond to the corresponding epidemiological trends and reduce the burden of them to some extent.

## Introduction

The history of research on cardiomyopathy can be traced back to a hundred years ago [1]. Cardiomyopathy comprises disorders characterized by structural-functional impairment of the myocardium, manifesting as diverse clinical presentations ranging from asymptomatic ventricular dysfunction and exercise-induced dyspnea to severe outcomes (heart failure, sudden death). This condition arises from multiple etiologies, including genetic predisposition, obesity, type 2 diabetes, hypertension, and excessive alcohol consumption [2–9]. AC is a cardiovascular disease caused by excessive alcohol consumption (typically defined as more than 80 grams per day for 5 years or longer) and belongs to an acquired dilated cardiomyopathy (DCM) [10,11] Although many studies have led to innovations and breakthroughs in the treatment of cardiomyopathy, limitations remain, which require traditional treatment options such as invasive surgery, like heart transplantation [2,5].

Previous studies have shown that in 2019, childhood myocarditis and cardiomyopathy affected 121,259 boys and 77,216 girls worldwide [12]. Another study, using the GBD2019 database, found an estimated 710,000 AC cases and 3.73 million OC cases across the entire global population [13] In 2021, cardiomyopathy and myocarditis ranked 9th in Central, Eastern, and Central Asia combined (7th in Eastern Europe) among the top ten Level 3 causes by DALYs rate [14]. A review in 2023

indicates that the annual sudden cardiac death (SCD) incidence rate due to cardiomyopathy is between 0.15% and 0.7% globally, significantly higher than that in other age and gender-matched populations [15]. In addition, endomyopathy (CMPs) due to cardiac damage and damage to cardiomyocytes, including exacerbation of underlying CMPs or emergence of new CMPs, was common among patients suffering from COVID-19 during the novel coronavirus pandemic [16]. Although many recent epidemiological studies of cardiomyopathy have been conducted in different regions or populations, and corresponding control measures have been proposed, the global burden of cardiomyopathy is still not optimistic [17–19]. Previous studies have also found a significant association between extreme temperature (high or low) and risk of cardiomyopathy and exacerbation of symptoms in patients with cardiomyopathy [20,21].

Despite the Global Burden of Disease (GBD) studies providing extensive data on various cardiovascular diseases, a comprehensive analysis specifically for cardiomyopathy (including AC, among others) as a comprehensive category is still lacking. Neither the overall trends of TC, AC, and OC diseases from 1990 to 2021 have been integrated, nor have future disease prediction studies been conducted. This study aims to fill this gap by systematically organizing the global, regional, and national burden of cardiomyopathy as an independent disease entity from 1990 to 2021 using data from the GBD database. It will analyze and compare the age-standardized incidence rates (ASPRs), age-standardized mortality rates (ASIRs), and age-standardized DALYs for TC, AC, and OC, and predict disease trends from 2022 to 2040 by dividing age into 20 five-year age groups (<5 years, 5–9 years, 10–14 years, …, 90–94 years, and ≥95 years) using the Bayesian age-period-cohort (BAPC) model. By synthesizing data scattered within the GBD framework, this study provides a unique integrated assessment, establishing a critical benchmark for understanding the total public health burden of all forms of cardiomyopathy. This is crucial for informed policy planning and resource allocation, which typically target disease groups rather than isolated ICD codes.

## Methods

### Data source and collection

GBD 2021 provides and systematically analyses comprehensive estimates of the global burden of 371 diseases and injuries, 288 causes of death, and 88 risk factors for 204 countries and territories from 1990 to 2021. These analyses were achieved by using sophisticated statistical models such as the Cause of Death Envelope Model (CODEm) framework, MR-BRT, and DisMod-MR 2.1 to adjust for bias due to differences in data sources, definitions, and measurement methods, ensuring internal consistency of estimates across regions, ages, genders and years [14,22,23].

This study used data on the disease burden of AC and OC from GBD 2021 from 1990 to 2021, obtained using the Global Health Data Exchange (GHDx) results tool. Epidemiological data related to AC and OC were established according to the 9th and 10th editions of the International Classification of Diseases (ICD-9 and ICD-10), where AC corresponds to the disease code 425.5 (ICD-9) or I42.6 (ICD-10), and OC corresponds to the codes 425.0–425.18, 425.3, and 425.8–425.9 (ICD-9) or I42.0–I42.5 and I42.7 (ICD-10). TC was evaluated by integrating data about AC and OC, with 95% confidence intervals calculated based on the 95% uncertainty interval (UI) width divided by the standardized error value, determined by 1.96 × 2.

In addition, the University of Washington Institutional Review Board waived the requirement for informed consent when accessing GBD data [24]. This study adhered to the Guidelines for Accurate and Transparent Health Estimates Reporting (GATHER) [25].

### Sociodemographic index

The socio-demographic index (SDI), developed by researchers from the Global Burden of Disease Study (GBD), uses data on fertility, education levels, and per capita income to quantify the level of development of a country or region. The indicator is divided into five levels (low, low-middle, middle, high-middle, and high), ranging from 0 to 1, and a higher SDI

indicates a greater level of socio-economic development [26]. This index was used in this study to explore the relationship between the burden of cardiomyopathy and socio-economic development levels.

## Statistical analysis

For statistical analysis of trends in prevalence, mortality, and DALYs for TC, AC, and OC, we need to calculate estimated annual percentage change (EAPC) using their corresponding ASRs. Its calculation formula is as follows:

$$ASR = \frac{\sum_{i=1}^{A} a_i w_i}{\sum_{i=1}^{A} w_i} \times 100,000$$

($a_i$: the age-specific rate in the ith age group; $w$: the number of people in the corresponding ith age group among the standard population; $A$: the number of age groups).

The required ASR is already included in the disease data obtained from GBD 2021. This study needs to use a regression model that describes a pattern of age-standardized ratios for a specific period to calculate the EAPC value [27]. The model equation is as follows:

$$Y = \alpha + \beta X + \varepsilon$$

Where $Y$ represents ln(ASR), $X$ represents the calendar year, $\alpha$ is the intercept term, $\beta$ represents the slope or trend, and $\varepsilon$ is the error term. The EAPC annual percentage change calculation formula can be expressed as:

$$EAPC = 100 \times (exp(\beta) - 1)$$

Similarly, the 95% confidence interval (CI) of EAPC can also be calculated from the above linear regression model. If the lower limit of EAPC and its 95% CI are > 0, the ASR is considered to have an increasing trend. Conversely, if the EAPC and its upper limit of 95% CI are both < 0, it is considered that ASR has a downward trend. If neither condition is met, the ASR is considered stable. This study used Spearman correlation to assess the correlation between different SDI level regions and cardiomyopathy ASPR, ASDR, and age-standardized DALYs.

The Bayesian age-period-cohort (BAPC) model demonstrates unique comprehensive advantages when processing the GBD database, with its core lying in adopting a hierarchical Bayesian framework to model global, regional, and national data as an integrated system. Through "information leverage," it significantly enhances the robustness of estimates for sparse data, such as small countries and rare diseases. The model can simultaneously decompose age, period, and cohort effects, providing in-depth attributional analysis for changes in disease burden. It leverages complete Bayesian inference to quantify uncertainty for all predictions comprehensively. Meanwhile, its inherent smooth prior ensures that trend predictions align with epidemiological common knowledge, avoiding unreasonable fluctuations. Ultimately, it generates estimates for all countries and diseases under unified model standards, ensuring the fairness and comparability of global disease burden rankings, making it a robust, scientifically sound, and practical tool for global public health policy making [28,29]. TheBAPC model was used in this study to predict the trend of cardiomyopathy from 2022 to 2040 by dividing age into 20 five-year age groups (<5 years, 5–9 years, 10–14 years,..., 90–94 years, and ≥95 years). The BAPC model was specified as:

$$\eta_{ap} = \mu + \alpha_a + \beta_p + \gamma c + \varepsilon_{ap}$$

Where $\eta_{ap}$ represents the log-rate for age group $a$ and period $p$, μ is the intercept, $\alpha_a$ denotes age effects, $\beta_p$ represents period effects, $\gamma c$ represents cohort effects (with cohort c = p − a), and $\varepsilon_{ap}$ is the error term. BAPC's usage comes from the R BAPC package.

All of the above data analysis processing calculations were performed using the R software (version 4.4.3), and all analysis results with P<0.05 were considered statistically significant.

## Ethics statement

Since the GBD study used publicly accessible data that did not contain any confidential or identifiable patient information, the Washington University Institutional Review Board reviewed and approved the exemption for informed consent. (https://www.healthdata.org/research-analysis/gbd).

## Results

### Global trends

**Prevalence.** Globally, TC, AC, and OC in 2021 were estimated to be 4752361.3 cases (95% UI: 3937775.4–5566947.2), 528429 cases (95% UI: 4,39582.3–639167.4), and 4223932.2 cases (95% UI: 3417357.4–5034257.6), all of which increased in prevalence compared with 1990, with percentage increases of 79.9%, 48.24%, and 84.83%, respectively. The ASPR per 100,000 persons for TC, AC, and OC were 59.5 (95% UI: 49.4–69.6), 6.2 (95% UI: 5.1–7.4), and 53.3 (95% UI: 43.6–63.8), respectively. Compared with 1990, the ASPR per 100,000 persons increased for both total cardiomyopathy and OC, with EAPCs of 0.16 (95% UI: 0.15–0.18) and 0.27 (95% UI: 0.25–0.29), respectively, but the ASPR per 100,000 persons for AC showed a decreasing trend, with an EAPC of −0.59 (95% UI: −0.83 to −0.35) (Table 1, S1 Table, S4 Table, Fig 1).

**Mortality.** Globally, TC, AC, and OC in 2021 were estimated to be 370273.6 cases, 64011.4 cases, and 306262.2 cases, all of which were elevated compared to the number of deaths in 1990, with percentage increases of 45.65%, 35.98%, and 47.85%, respectively. The ASDR per 100,000 persons for TC, AC, and OC were 4.5, 0.7, and 3.7, respectively. Compared with 1990, the ASDR per 100,000 persons for TC, AC, and OC all decreased, with EAPCs of −1.63, −1.72, and −1.62 (Table 2, S2 Table, S5 Table, Fig 1).

### Disability-adjusted life years

Globally, the estimated DALYs for TC, AC, and OC in 2021 were 10690818.8, 2185527.6 and 8505291.2, all increased compared to 1990 DALYs, with percentage increases of 41.76%, 39.94%, and 42.23%, respectively. The ASR_DALYs per 100,000 persons for TC, AC, and OC were 129.8, 25.3, and 104.5, respectively. Compared with 1990, ASR_DALYs per 100,000 persons decreased for TC, AC, and OC, with EAPCs of −1.07, −1.44, and −0.97 (Table 3, S3 Table, S6 Table, Fig 1).

### Socio-demographic index regions

**Total cardiomyopathy.** For TC in 2021, ASPR per 100,000 persons peaked in High SDI regions (107.5), while Middle SDI regions showed the lowest (40.1), though Low SDI regions (74.9) ranked second only to High SDI. EAPC for ASPR per 100,000 persons (−0.23) remained negative exclusively in High SDI regions, contrasting with positive EAPC values across other SDI strata. ASDR (6.6) and ASR_DALYs (204.4) per 100,000 persons both peaked in High-middle SDI regions; Low SDI regions reported second-highest values (ASDR 6.5; ASR_DALYs 180.8), whereas Middle SDI regions recorded minima (ASDR 2.8; ASR_DALYs 77.4). Notably, all three metrics exhibited minimum EAPC values in High SDI regions: ASDR per 100,000 persons (−2.68) and ASR_DALYs per 100,000 persons (−2.5) (Table 1–Table 3, Fig 1A). In 2021, TC cases demonstrated younger age distributions in lower SDI regions but shifted to older distributions in higher SDI regions, with age-related disparity widening as age groups advanced (Fig 2A). What' more, ASPR, ASDR, and ASR_DALYs of TC showed negative correlations with SDI at 0.5, shifted to positive correlations at 0.75, and reverted to negative correlations with further SDI increases (Fig 3).

**Table 1. 1990–2021 Global and regional prevalence trends in total cardiomyopathy burden.**

| location | Total Cardiomyopathy Prevalence (95% UI) | | | | |
|---|---|---|---|---|---|
| | Number_1990 | ASR per 100,000_1990 | Number_2021 | ASR per 100,000_2021 | EAPC_95% CI |
| Global | 2641734.7 (2242999.5–3040469.9) | 57.5 (49.1–65.9) | 4752361.3 (3937775.4–5566947.2) | 59.5 (49.4–69.6) | 0.16 (0.15–0.18) |
| High SDI | 1065034.8 (912775.6–1217294) | 116.6 (100.6–132.6) | 1448054.1 (1211937.4–1684170.9) | 107.5 (91.3–123.8) | −0.23 (−0.26 to −0.21) |
| High-middle SDI | 541598.1 (469519–613677.3) | 56.6 (48.5–64.7) | 859609.3 (727513.8–991704.7) | 58.8 (50.1–67.5) | 0.27 (0.22–0.33) |
| Middle SDI | 424884.9 (347858.7–501911.1) | 29.8 (24.9–34.7) | 952037.3 (769074–1135000.5) | 40.1 (32.7–47.5) | 1.04 (0.95–1.12) |
| Low-middle SDI | 357429.6 (290946.5–423912.8) | 39.7 (33–46.4) | 825217.2 (651492.1–998942.4) | 49 (39.4–58.6) | 0.66 (0.64–0.69) |
| Low SDI | 249561.6 (188805.7–310317.4) | 64.2 (49.6–78.8) | 661916 (501148.2–822683.8) | 74.9 (57.7–92.2) | 0.52 (0.5–0.53) |
| Andean Latin America | 10665.2 (8415.8–12914.6) | 30.2 (24.6–35.8) | 21897.6 (17313.3–26481.8) | 34.4 (27.4–41.5) | 0.58 (0.43–0.73) |
| Australasia | 24329.9 (20751.3–27908.5) | 120 (102.2–137.8) | 48266.7 (40623.4–55909.9) | 130.2 (108.4–152) | 0.54 (0.4–0.67) |
| Caribbean | 16648.7 (13749.6–19547.8) | 50.2 (41.7–58.6) | 35419.5 (30080.8–40758.1) | 72 (61–83) | 1.36 (1.28–1.44) |
| Central Asia | 20808.9 (17130.6–24487.2) | 31.9 (26.9–36.9) | 56642.7 (47216.5–66068.8) | 63.3 (52.7–73.9) | 2.95 (2.7–3.2) |
| Central Europe | 119930.2 (100631.6–139228.8) | 95.1 (80–110.2) | 197756.6 (160987.5–234525.7) | 118.1 (99.7–136.6) | 0.95 (0.8–1.09) |
| Central Latin America | 60432 (48625.7–72238.2) | 37.6 (31.4–43.7) | 103546.7 (83865.3–123228) | 42.2 (34.3–50.2) | 0.37 (0.3–0.44) |
| Central Sub-Saharan Africa | 31054.5 (21950.4–40158.5) | 72.5 (52.2–92.8) | 87713.5 (61109–114318) | 80.2 (58.2–102.3) | 0.31 (0.28–0.33) |
| East Asia | 106692.4 (84299.3–129085.5) | 10 (8.1–11.9) | 300085 (236306.1–363863.9) | 19.7 (15.5–24) | 2.73 (2.44–3.03) |
| Eastern Europe | 219806.7 (190015.3–249598.1) | 94.8 (82.2–107.4) | 323184.5 (277536.5–368832.5) | 136 (116.9–155) | 1.45 (1.32–1.59) |
| Eastern Sub-Saharan Africa | 158254.2 (119410–197098.4) | 111.9 (84–139.8) | 402362.8 (307368.7–497357) | 126.7 (94.9–158.5) | 0.31 (0.28–0.34) |
| High-income Asia Pacific | 145218.4 (121835.2–168601.6) | 83.6 (69.2–98) | 207491 (166585.4–248396.5) | 84.3 (69.9–98.7) | 0.23 (0.16–0.3) |
| High-income North America | 475379.1 (400685.8–550072.4) | 162.9 (137.7–188) | 579544.2 (483536.9–675551.6) | 135.4 (113.3–157.5) | −0.79 (−0.85 to −0.74) |
| North Africa and Middle East | 108680.1 (85520–131840.1) | 30.5 (25.1–35.8) | 225005.7 (176441.3–273570.1) | 37.5 (29.9–45.1) | 0.69 (0.64–0.75) |
| Oceania | 1050.5 (806.9–1294.2) | 20.9 (16.6–25.3) | 2706.6 (2047.3–3365.8) | 24.2 (18.4–29.9) | 0.52 (0.49–0.56) |
| South Asia | 279720.1 (228209.4–331230.8) | 34 (28.1–39.8) | 718523.1 (573392.8–863653.4) | 44.8 (36.3–53.4) | 1.05 (0.99–1.11) |
| Southeast Asia | 73091.5 (58987.9–87195.2) | 23.5 (19.2–27.8) | 164367.4 (130971.2–197763.5) | 26.8 (21.5–32.2) | 0.28 (0.24–0.33) |
| Southern Latin America | 49050.3 (40705.4–57395.2) | 102.8 (84.6–121) | 73399.4 (58930.2–87868.7) | 99.7 (81.3–118.2) | −0.15 (−0.25 to −0.06) |
| Southern Sub-Saharan Africa | 33160 (24481.4–41838.5) | 75.4 (56.6–94.2) | 59624.9 (44507.5–74742.4) | 79.9 (59.6–100.2) | 0.11 (0.02–0.2) |
| Tropical Latin America | 144439.9 (121194.1–167685.8) | 121 (101.1–141) | 284686.2 (233197.1–336175.3) | 122.7 (99.9–145.5) | −0.15 (−0.23 to −0.07) |
| Western Europe | 465077.8 (389851.4–540304.2) | 105.9 (90.9–120.8) | 594168.1 (485269.9–703066.3) | 96.5 (82–111.1) | −0.22 (−0.28 to −0.16) |
| Western Sub-Saharan Africa | 98244.3 (72557.9–123930.7) | 67.5 (50.2–84.8) | 265969.2 (191434.1–340504.4) | 71.5 (52.3–90.8) | 0.14 (0.12–0.16) |

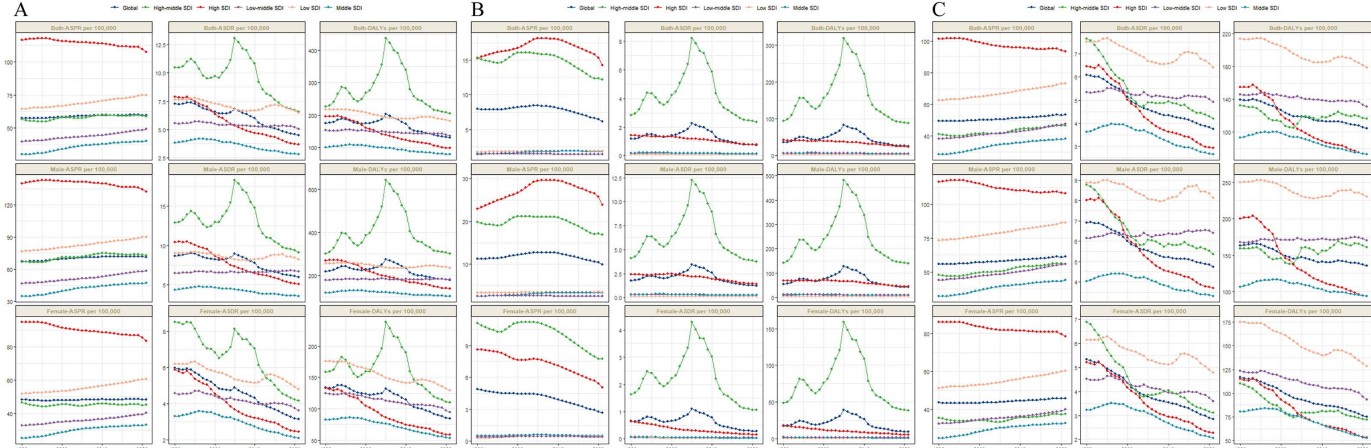

**Fig 1. Trends in prevalence, incidence, deaths, and disability-adjusted life-years from 1990 to 2021.** (A) Total cardiomyopathy. (B) Alcoholic cardiomyopathy. (C) Other cardiomyopathy.

## Alcoholic cardiomyopathy

In 2021, ASPR per 100,000 persons for AC peaked in High SDI regions (14.2), followed by High-middle SDI regions; Low-middle SDI regions recorded the lowest (1.4). Highest ASDR (2.4) and ASR_DALYs (88.3) per 100,000 persons were both observed in High-middle SDI regions. Minimum EAPC for ASPR (−0.58) occurred in High-middle SDI regions, whereas ASDR and ASR_DALYS showed minima in High SDI regions (−2.08 and −1.74) (S1–S3 Tables, Fig 1B). In 2021, AC cases were predominantly distributed in High SDI regions and High-middle SDI regions, with age-related concentration increasing among middle-aged and elderly populations; male patients outnumbered females by notable margins (S1 Fig). For AC's ASPR, ASDR, and ASR_DALYs, we observed a non-significant association with SDI until 0.6, followed by a strong positive correlation at 0.75 that shifted to negative with further SDI increases (S3 Fig).

## Other cardiomyopathy

In 2021, ASPR per 100,000 persons for OC peaked in High SDI regions (93.3), followed by Low SDI regions (73). ASDR (6.4) and ASR_DALYs (178.2) per 100,000 persons both peaked in Low SDI regions. Notably, Middle SDI regions exhibited near-minimal values: ASPR (38.3) and ASDR (2.6), with ASR_DALYs at 72.8—slightly higher than High SDI regions' 72.7. Minimum EAPC values for all three indicators occurred in High SDI regions: ASPR (−0.29), ASDR (−2.82), and ASR_DALYs (−2.73) (S4–S6 Tables, Fig 1C). In 2021, OC cases predominantly involved younger populations concentrated in lower SDI regions, contrasting with TC's similar elderly predominance in higher SDI regions; yet male middle-aged/elderly for OC showed notable fewer cases compared to TC, whereas female distributions exhibited no significant difference (S2 Fig). For the ASPR, ASDR, and ASR_DALYs of OC, we observed a negative correlation with SDI at 0.5, which then turns into a sustained positive correlation until 0.75, after which the ASDR and ASR_DALYs metrics turn back to a negative trend as SDI rises further (S4 Fig).

## 21 GBD regions

From the results of the study, it is easy to see that the disease trends of TC, AC, and OC had significant differences among 21 GBD regions in 2021.

**Table 2. 1990–2021 Global and regional mortality trends in total cardiomyopathy burden.**

| location | Total Cardiomyopathy Deaths (95% UI) | | | | |
|---|---|---|---|---|---|
| | Number_1990 | ASR per 100,000_1990 | Number_2021 | ASR per 100,000_2021 | EAPC_95% CI |
| Global | 254222.2 (230367.4–278077) | 7.3 (6.6–7.9) | 370273.6 (340965.8–399581.3) | 4.5 (4.1–4.8) | −1.63 (−1.81 to −1.45) |
| High SDI | 84440.6 (80107.9–88773.3) | 7.9 (7.5–8.3) | 80709.5 (73718.9–87700.2) | 3.7 (3.4–3.9) | −2.68 (−2.78 to −2.57) |
| High-middle SDI | 81962.7 (77354–86571.5) | 10.5 (9.7–11.2) | 119684.4 (111748.5–127620.4) | 6.6 (6.1–7) | −1.47 (−2 to −0.93) |
| Middle SDI | 35158.9 (29924–40393.8) | 3.8 (3.3–4.4) | 67447.4 (61573.4–73321.3) | 2.8 (2.5–3) | −1.31 (−1.5 to −1.12) |
| Low-middle SDI | 33596.9 (23365.6–43828.2) | 5.5 (3.8–7.3) | 67734 (56333.2–79134.9) | 5.1 (4.2–6) | −0.27 (−0.32 to −0.21) |
| Low SDI | 18584.6 (12794–24375.3) | 7.6 (4.8–10.5) | 34113 (24726.4–43499.6) | 6.5 (4.6–8.3) | −0.47 (−0.6 to −0.34) |
| Andean Latin America | 518.9 (442.9–594.9) | 2 (1.7–2.3) | 645 (522.5–767.5) | 1.1 (0.9–1.3) | −1.79 (−2.15 to −1.42) |
| Australasia | 1585.8 (1492.9–1678.7) | 7.2 (6.8–7.6) | 1656.7 (1507.2–1806.2) | 3.1 (2.8–3.3) | −2.42 (−2.86 to −1.98) |
| Caribbean | 1231.7 (1010.6–1452.7) | 4.5 (3.7–5.3) | 2951.5 (2516–3386.9) | 5.6 (4.7–6.5) | 1.09 (0.88–1.31) |
| Central Asia | 2194.4 (1912.2–2476.7) | 4.5 (3.9–5.1) | 10887.5 (9172.8–12602.1) | 13 (11.1–14.9) | 4.64 (3.55–5.74) |
| Central Europe | 22385.8 (20931.9–23839.8) | 17.6 (16.4–18.8) | 27188.7 (24842.7–29534.6) | 12.1 (11–13.2) | −1.28 (−1.54 to −1.02) |
| Central Latin America | 2175 (2073.1–2276.9) | 2.4 (2.2–2.5) | 4262.8 (3765.8–4759.8) | 1.8 (1.5–2) | −1.48 (−1.66 to −1.3) |
| Central Sub-Saharan Africa | 3627.1 (2128.8–5125.5) | 15.7 (8–23.4) | 7529.2 (3661.5–11396.9) | 14.2 (6.3–22.1) | −0.32 (−0.35 to −0.29) |
| East Asia | 7077 (3667.5–10486.5) | 0.9 (0.4–1.3) | 17439.7 (12990.8–21888.6) | 1 (0.7–1.2) | 0.4 (0.06–0.75) |
| Eastern Europe | 30455.5 (28638.8–32272.2) | 12.2 (11.5–12.9) | 69697.7 (64590–74805.5) | 23.3 (21.6–25) | 1.88 (0.69–3.09) |
| Eastern Sub-Saharan Africa | 5024 (3627.9–6420.1) | 4.7 (3.5–5.9) | 8818.6 (6107.6–11529.6) | 4 (2.9–5.1) | −0.59 (−0.68 to −0.5) |
| High-income Asia Pacific | 10532.3 (9761.9–11302.7) | 6.1 (5.6–6.6) | 9990.2 (8540.6–11439.8) | 1.8 (1.6–2) | −3.29 (−3.88 to −2.7) |
| High-income North America | 30154.2 (28619–31689.3) | 8.7 (8.3–9.1) | 30262 (27889.1–32634.9) | 4.7 (4.4–5.1) | −2.55 (−2.74 to −2.35) |
| North Africa and Middle East | 6745.3 (4646.1–8844.5) | 3 (1.7–4.3) | 8788.9 (6140.3–11437.4) | 2 (1.3–2.7) | −1.27 (−1.33 to −1.22) |
| Oceania | 150.9 (92.2–209.5) | 4.2 (2.5–5.9) | 379.9 (231.5–528.3) | 4.2 (2.5–5.9) | 0.06 (0.02–0.1) |
| South Asia | 27341.1 (15413.3–39268.8) | 4.8 (2.6–7) | 67758.2 (51893.1–83623.2) | 5 (3.9–6.2) | 0.36 (0.25–0.48) |
| Southeast Asia | 8249.9 (6499.1–10000.7) | 3.9 (3.1–4.8) | 17143.3 (14539–19747.6) | 3.2 (2.7–3.6) | −0.99 (−1.18 to −0.8) |
| Southern Latin America | 6680.3 (6095.6–7265) | 15.5 (14.1–16.9) | 7429.7 (6754.7–8104.6) | 8.3 (7.6–9.1) | −2.07 (−2.25 to −1.88) |
| Southern Sub-Saharan Africa | 3754.3 (2987–4521.5) | 15.1 (11.6–18.6) | 6524.8 (5783.1–7266.5) | 12.8 (11.3–14.2) | −0.64 (−0.93 to −0.35) |
| Tropical Latin America | 13390 (12826.6–13953.3) | 15.6 (14.8–16.5) | 17353 (16070.4–18635.6) | 6.9 (6.4–7.5) | −3.23 (−3.52 to −2.93) |
| Western Europe | 58911.2 (54467–63355.3) | 10.4 (9.6–11.1) | 35954.8 (32311.8–39597.8) | 3.4 (3.1–3.7) | −3.92 (−4.23 to −3.61) |
| Western Sub-Saharan Africa | 12037.6 (8504.9–15570.2) | 13.9 (9.5–18.3) | 17611.5 (13804.8–21418.3) | 8.7 (7–10.3) | −1.79 (−1.93 to −1.66) |

## Total cardiomyopathy

For TC, in 2021, among 21 GBD regions, regions with higher TC ASPR per 100,000 persons included Eastern Europe (136), High-income North America (135.4), Australasia (130.2), and Eastern Sub-Saharan Africa (126.7). EAPC for ASPR per 100,000 persons was negative in only four regions, with the maximum value of 2.95 in Central Asia and the minimum value of −0.79 in High-income North America. TC had higher ASDR per 100,000 persons in Eastern Europe (23.3), Central Sub-Saharan Africa (14.2), Southern Sub-Saharan Africa (12.8), and Central Europe (12.1). EAPC for ASDR per 100,000 persons was positive in six regions and negative in the rest, with the maximum value of 4.64 in Central Asia and the minimum value of −3.92 in Western Europe. Additionally, TC had the largest ASR_DALYs per 100,000 persons in Eastern Europe (864.2), and EAPC for ASR_DALYs per 100,000 persons was positive in five regions and negative in the

**Table 3. 1990–2021 Global and regional DALYs trends in total cardiomyopathy burden.**

| location | Total Cardiomyopathy DALYs (95% UI) | | | | |
|---|---|---|---|---|---|
| | Number_1990 | ASR per 100,000_1990 | Number_2021 | ASR per 100,000_2021 | EAPC_95% CI |
| Global | 7541688.4 (6684606.4–8398770.5) | 175.1 (157.8–192.4) | 10690818.8 (9864323.4–11517314.2) | 129.8 (119.7–140) | −1.07 (−1.37 to −0.76) |
| High SDI | 1990081.7 (1922446.5–2057716.9) | 196.3 (189.7–203) | 1669168.3 (1568982.4–1769354.2) | 97 (91.8–102.3) | −2.5 (−2.6 to −2.41) |
| High-middle SDI | 2109501.8 (2001931.2–2217072.4) | 225.9 (213.4–238.3) | 3511891.8 (3285172.2–3738611.4) | 204.4 (191.6–217.3) | −0.43 (−1.24 to 0.39) |
| Middle SDI | 1286944.1 (1080708.5–1493179.6) | 100.1 (85.4–114.8) | 1960925.6 (1792691.2–2129160.1) | 77.4 (71–83.9) | −1.07 (−1.22 to −0.92) |
| Low-middle SDI | 1322396.6 (922910.5–1721882.6) | 152.1 (107.3–196.9) | 2159235.9 (1821876.6–2496595.1) | 135.7 (114.3–157) | −0.33 (−0.37 to −0.3) |
| Low SDI | 821628.9 (594886–1048371.9) | 216.6 (150.1–283.2) | 1375930.4 (1018126.2–1733734.7) | 180.8 (133–228.6) | −0.59 (−0.68 to −0.49) |
| Andean Latin America | 26399.7 (20933.6–31865.8) | 71.7 (60.2–83.2) | 23695.3 (19320.5–28070) | 38.1 (31–45.1) | −1.81 (−2.17 to −1.45) |
| Australasia | 41535.6 (39504.6–43566.5) | 190.8 (181.6–199.9) | 38791.9 (36038.2–41545.5) | 88.1 (81.7–94.6) | −2.16 (−2.57 to −1.76) |
| Caribbean | 47803.3 (35100.5–60506.1) | 151.6 (117.7–185.5) | 91041.8 (73173.2–108910.4) | 180.8 (141.8–219.8) | 0.96 (0.76–1.16) |
| Central Asia | 79366.9 (69767.4–88966.4) | 141.6 (123.9–159.3) | 359966.1 (301501.7–418430.5) | 386.1 (325.6–446.5) | 4.34 (3.2–5.49) |
| Central Europe | 459793.1 (433966.2–485620) | 340.1 (321–359.1) | 536149.8 (485549.7–586749.8) | 268 (241.3–294.7) | −0.79 (−0.95 to −0.63) |
| Central Latin America | 92469.2 (87637.3–97301.2) | 70.9 (67.8–74.1) | 138930.2 (122284.5–155575.8) | 56.5 (49.4–63.6) | −1.03 (−1.15 to −0.91) |
| Central Sub-Saharan Africa | 157250.9 (102616.7–211885.1) | 417.3 (240.6–594.1) | 285028.7 (150953.3–419104.2) | 361.4 (177–545.8) | −0.44 (−0.48 to −0.41) |
| East Asia | 343948.3 (200681.7–487214.9) | 32.5 (18.8–46.3) | 475009.8 (368812.6–581207.1) | 28.2 (22.3–34.1) | −0.63 (−0.89 to −0.38) |
| Eastern Europe | 1058289.5 (989083.8–1127495.2) | 416 (389.6–442.4) | 2377494.5 (2203109.2–2551879.9) | 864.2 (803–925.5) | 1.97 (0.67–3.28) |
| Eastern Sub-Saharan Africa | 280297.1 (192018.3–368575.8) | 169.8 (126.3–213.4) | 443195.4 (302142.8–584247.9) | 142.7 (101.5–183.8) | −0.57 (−0.65 to −0.5) |
| High-income Asia Pacific | 255970.7 (242395.5–269545.9) | 141.4 (133–149.8) | 177106.4 (159462.2–194750.6) | 49.3 (45.3–53.2) | −3.07 (−3.45 to −2.68) |
| High-income North America | 799841.7 (770994.3–828689) | 249.4 (240.7–258) | 697089 (662671.2–731506.7) | 132.1 (125.7–138.5) | −2.56 (−2.74 to −2.38) |
| North Africa and Middle East | 390044.9 (227333.4–552756.5) | 112 (76.7–147.2) | 349328.7 (270034.8–428622.5) | 64.7 (48.5–80.8) | −1.7 (−1.77 to −1.64) |
| Oceania | 7448.7 (4418.1–10479.2) | 139.2 (85.7–192.8) | 18288.6 (11463.6–25113.6) | 147.5 (92–203) | 0.26 (0.22–0.3) |
| South Asia | 1046248.1 (613893.7–1478602.5) | 133.2 (76.8–189.5) | 2038330.7 (1565071.9–2511589.6) | 130.9 (100.7–161) | 0.09 (0.02–0.16) |
| Southeast Asia | 253196.7 (200020.5–306372.9) | 85.4 (67.6–103.2) | 474712.8 (391648.3–557777.4) | 73.7 (61.8–85.6) | −0.72 (−0.85 to −0.58) |
| Southern Latin America | 165960 (153648–178272) | 359.5 (332.2–386.8) | 154446.8 (143937.1–164956.4) | 186.3 (173.7–198.9) | −2.25 (−2.46 to −2.04) |
| Southern Sub-Saharan Africa | 112147.1 (96350.3–127943.9) | 341.4 (277.6–405.1) | 188419.5 (167318.6–209520.3) | 301.9 (268.2–335.7) | −0.44 (−0.64 to −0.23) |

*(Continued)*

**Table 3.** (Continued)

| location | Total Cardiomyopathy DALYs (95% UI) | | | | |
|---|---|---|---|---|---|
| | Number_1990 | ASR per 100,000_1990 | Number_2021 | ASR per 100,000_2021 | EAPC_95% CI |
| Tropical Latin America | 418325.8 (403123.2–433528.3) | 394.2 (379.3–409.1) | 484332.7 (459528.6–509136.7) | 193.8 (183.9–203.7) | −2.84 (−3.12 to −2.57) |
| Western Europe | 1055547.2 (999574.8–1111519.5) | 201.5 (191.3–211.7) | 635347.9 (586453.2–684242.7) | 77.8 (72.6–83) | −3.36 (−3.6 to −3.12) |
| Western Sub-Saharan Africa | 449804 (326575.7–573032.3) | 347.6 (244.5–450.6) | 704112.3 (533489–874735.6) | 225.3 (177.5–273.1) | −1.64 (−1.78 to −1.51) |

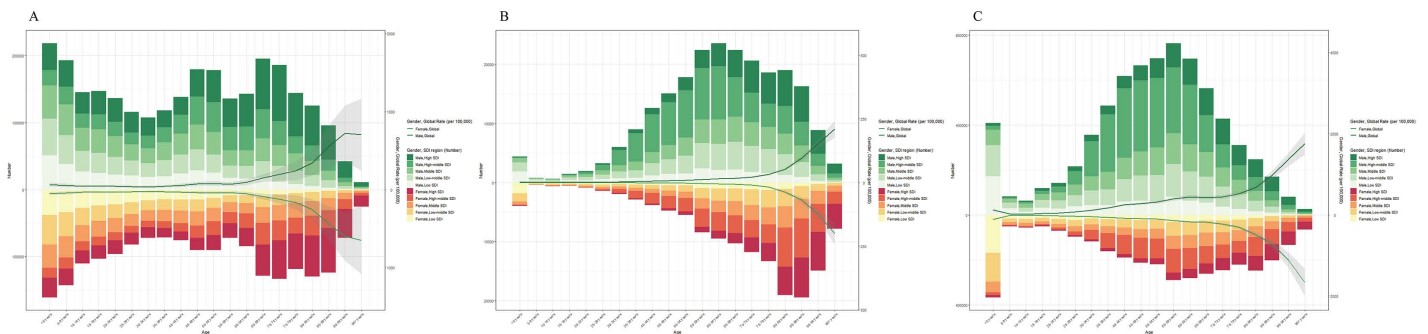

**Fig 2. The age-specific numbers and ASRs of total cardiomyopathy by SDI regions in 2021.** (A) ASPRs. (B) ASDRs. (C) ASR_DALYs.

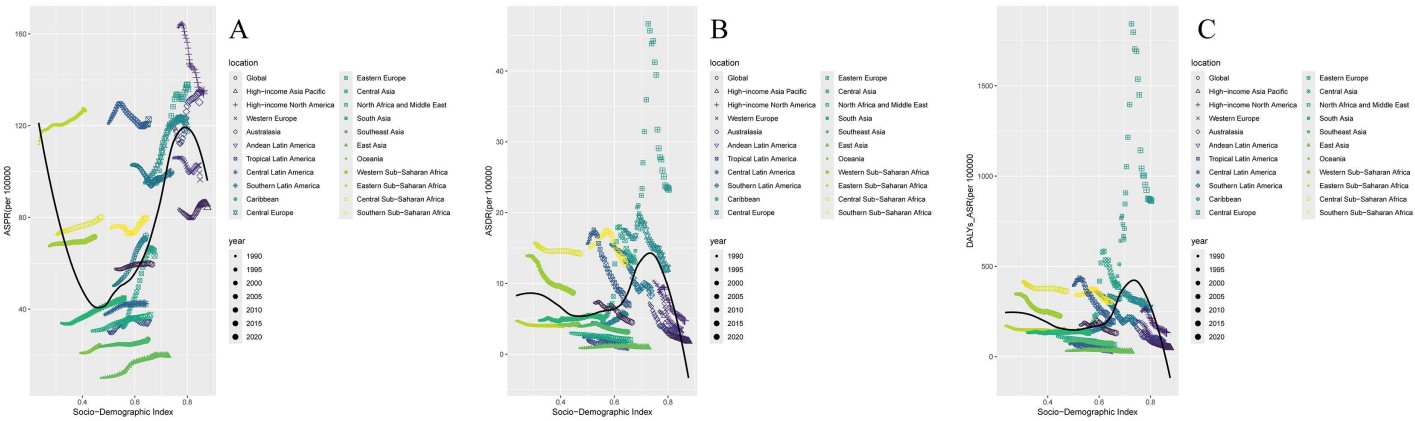

**Fig 3. Socio-demographic index trends with ASRs of total cardiomyopathy in 21 GBD regions in 2021.** (A) ASPRs. (B) ASDRs. (C) ASR_DALYs.

rest, with the maximum value of 4.34 in Central Asia and the minimum value of −3.36 in Western Europe. It is worth noting that TC had the highest ASPR per 100,000 persons, ASDR per 100,000 persons, and ASR_DALYs per 100,000 persons in Eastern Europe. In contrast, TC had the lowest ASPR per 100,000 persons (19.7), ASDR per 100,000 persons (1.0), and ASR_DALYs per 100,000 persons (28.2) in East Asia. Additionally, Central Asia had the highest EAPC for the above

three indicators of TC, while Western Europe had the lowest EAPC for ASDR per 100,000 persons and ASR_DALYs of TC (Table 1–Table 3 , Fig 3).

### Alcoholic cardiomyopathy

According to the research results, it can be seen that, similar to TC, AC had the highest ASPR per 100,000 persons (62.5), ASDR per 100,000 persons (13.1), and ASR_DALYs per 100,000 persons (510.5) in Eastern Europe. On the other hand, AC had the lowest ASPR per 100,000 persons (0), ASDR per 100,000 persons (0), and ASR_DALYs per 100,000 persons (0.1) in Andean Latin America, with the lowest ASDR per 100,000 persons also found in Central Sub-Saharan Africa (0), Eastern Sub-Saharan Africa (0), Southern Sub-Saharan Africa (0), and North Africa and Middle East (0). The EAPCs for ASPR per 100,000 persons were negative in eight regions and positive in the rest; the EAPCs for ASDR per 100,000 persons were positive in five regions and negative in the rest; the EAPCs for ASR_DALYs per 100,000 persons were positive in five regions and negative in the rest. Notably, the EAPC values for the three indicators were the highest in the Caribbean, with values of 5.65, 5.28, and −6.69, respectively; the EAPC values for the three indicators were the lowest in Southern Latin America, with values of −3.88, −6.69, and −6.72 (S1–S3 Tables, S3 Fig).

### Other cardiomyopathy

For OC, in 2021, Eastern Sub-Saharan Africa had the highest ASPR per 100,000 persons (124.1), and Central Sub-Saharan Africa had the highest ASDR per 100,000 persons (14.2) and ASR_DALYs per 100,000 persons (361.2). Notably, like TC, East Asia had the lowest ASPR per 100,000 persons (18.1), ASDR per 100,000 persons (0.9), and ASR_DALYs per 100,000 persons (24.6). Additionally, for OC, the EAPCs for ASPR per 100,000 persons were negative in only two regions and positive in the rest; the EAPCs for ASDR per 100,000 persons were positive in five regions and negative in the rest; the EAPCs for ASR_DALYs per 100,000 persons were positive in four regions and negative in the rest. Among these, High-income North America had the lowest EAPC for ASPR per 100,000 persons (−0.84); Central Asia had the highest EAPC for ASPR per 100,000 persons (3.15) and the highest EAPC for ASDR per 100,000 persons (5.09); Western Europe had the lowest EAPC for ASDR per 100,000 persons (−4.05) and the lowest EAPC for ASR_DALYs per 100,000 persons (−3.45); Eastern Europe had the highest EAPC for ASR_DALYs per 100,000 persons (5.39) (S4–S6 Tables, S4 Fig).

Additionally, this study analyzed the global and regional (5 SDI/21 GBD) distributions of AC/OC-attributable prevalence, deaths, and DALYs for TC in 1990 and 2021. Globally, OC accounted for >50% of cardiomyopathy-related prevalence, deaths, and DALYs, with stable proportions over time except in Eastern Europe, where OC prevalence/division ratios diverged significantly from global patterns. In Eastern Europe, AC prevalence/division ratios exceeded OC for all three indicators (Fig 4).

### National trends

**Total cardiomyopathy.** In 2021, global TC incidence (ASPR) ranged 8.3–188.6/100,000, with Montenegro (188.6), Ethiopia (182.9), and Poland (182.5) reporting highest rates versus Cook Islands (8.3), Jordan (14.2), and Thailand (15.2) (S7 Table, Fig 5A, Fig 6A). Mortality (ASDR) followed similar patterns (0.4–39.3/100,000), peaking in Kazakhstan (39.3), Latvia (27.9), and Montenegro (27.7) while Cook Islands (0.4), Jordan (0.4) and Thailand (0.7) remained lowest (S8 Table, Fig 5B, Fig 6B). DALYs (ASR_DALYs) showed steeper gradients (13.2–981.6/100,000), with Latvia (981.6), the Russian Federation (966.1), and Ukraine (727.2) reporting extremes, contrasting with Jordan (13.2), Cook Islands (13.5), and Kuwait (19.9) (S9 Table, Fig 5C, Fig 6C). Notably, Cook Islands, Jordan, and Thailand consistently presented the lowest values across all indicators, while extreme maxima predominantly occurred in high SDI nations, except Ethiopia (981.6) (Fig 6).

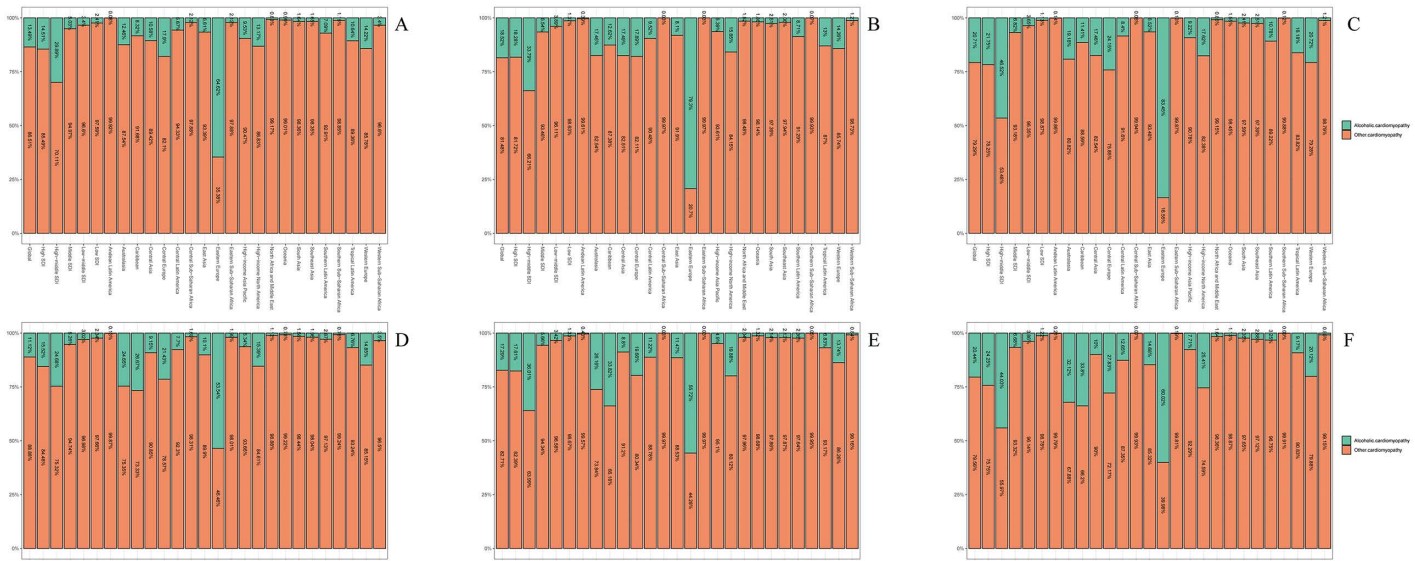

**Fig 4. Contribution of alcoholic cardiomyopathy and other cardiomyopathy to total cardiomyopathy prevalence, deaths, and DALYs, both sexes, globally and by region, in 1990 and 2021.** (A) 1990 prevalence. (B) 1990 deaths. (C) 1990 DALYs. (D) 2021 prevalence. (E) 2021 deaths. (F) 2021 DALYs.

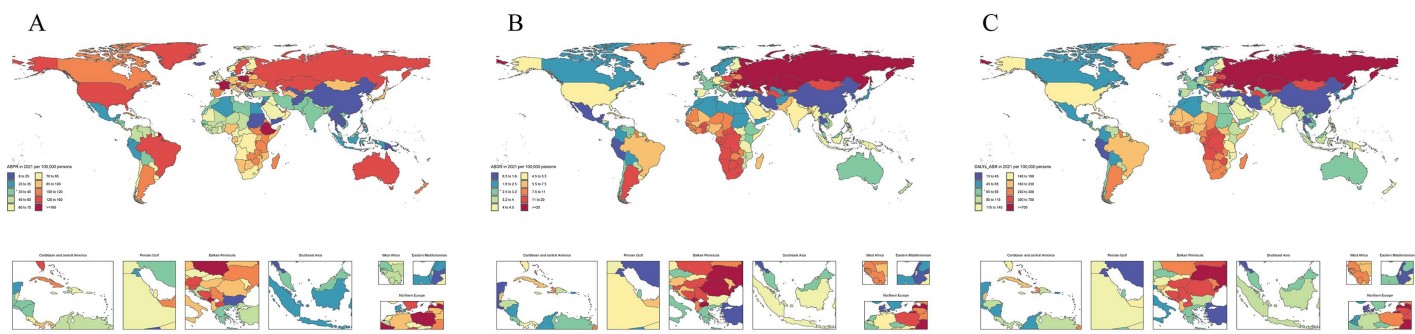

**Fig 5. ASRs of total cardiomyopathy in 204 countries and territories in 2021.** (A) ASPRs. (B) ASDRs. (C) ASR_DALYs.

## Alcoholic cardiomyopathy

In 2021, ASPR per 100,000 persons for AC globally ranged 0–65.7, with highest values in Russian Federation (65.7), Latvia (62.2), and Ukraine (60.9); lowest values remained 0 across seven countries including Ecuador and Tajikistan (S10 Table, S5A Fig, S7A Fig). Corresponding ASDR per 100,000 persons ranged 0–14.5, peaking at 14.5 in Latvia, 13.9 in Russian Federation, and 13.6 in Ukraine, while 43 countries (including Ecuador/Tajikistan) reported 0 (S11 Table, S5B Fig, S7B Fig). ASR_DALYs per 100,000 persons spanned 0–575.3, with Latvia (575.3), Russian Federation (542.4), and Ukraine (521.2) showing maxima; four countries (Cook Islands, Tajikistan, Egypt, Malaysia) recorded minima at 0, 0, 0, and 0 respectively (S12 Table, S5C Fig, S7C Fig). Notably, all three indicators' extreme maxima consistently occurred in the Russian Federation, Latvia, and Ukraine, with values rising proportionally to SDI elevation (S7 Fig).

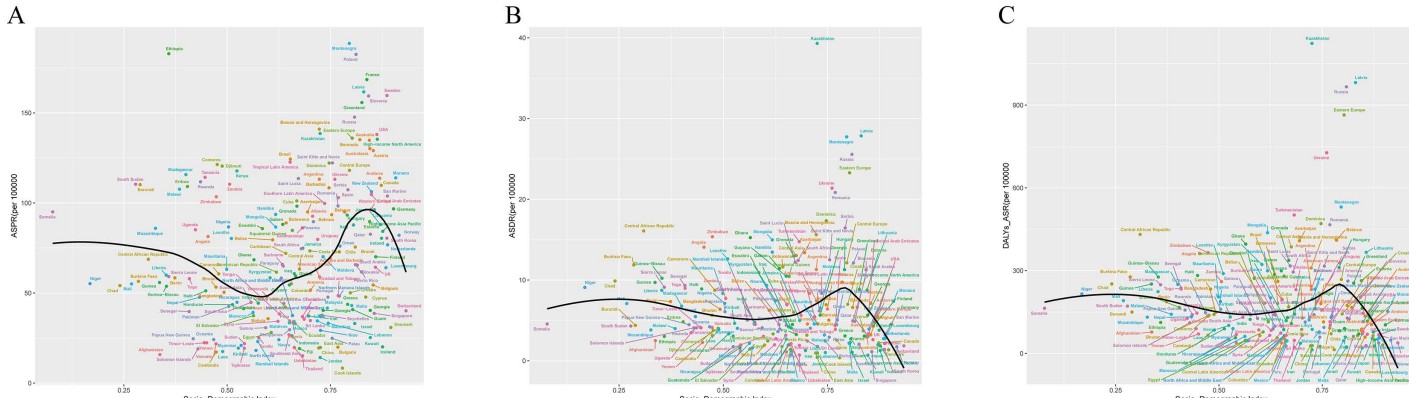

**Fig 6. Socio-demographic index trends with ASRs of total cardiomyopathy in 204 countries and territories in 2021.** (A) ASPRs. (B) ASDRs. (C) ASR_DALYs.

## Other cardiomyopathy

In 2021, ASPR per 100,000 persons for OC globally ranged 13.7–179.9, peaking in Ethiopia (179.9), Montenegro (159.8), and France (148); lowest values remained 13.7 in Kyrgyzstan, 14.1 in Jordan, and 14.6 in Thailand (S13 Table, S6A Fig, S8A Fig). Corresponding ASDR per 100,000 persons ranged 0.4–38.2, with Kazakhstan (38.2), Montenegro (23.5), and North Macedonia (20.5) reporting maxima; Jordan (0.4), Cook Islands (0.4), and Kuwait (0.6) showed minima (S14 Table, S6B Fig, S8B Fig). ASR_DALYs per 100,000 persons spanned 13.1–1083.3, peaking in Kazakhstan (1083.3), Turkmenistan (480.1), and Dominica (438.6); Jordan (13.1), Cook Islands (13.5), and Kuwait (16.5) maintained consistent minima across ASDR and ASR_DALYs (S15 Table, S6C Fig, S8C Fig). Excluding outlier nations (e.g., Kazakhstan, Ethiopia, and Montenegro), these OC metrics exhibited no significant associations with SDI trends.

## Age and sex

In 2021, ASPR for TC remained stable in 0–59 age groups but increased with age (peaking in older cohorts), while ASDR and ASR_DALYs consistently rose across all > 5 age strata. Males exhibited pronounced dominance over females in all three indicators within the same age groups, except females predominated in ≥80 age groups (Fig 2). For AC, males showed significantly higher prevalence, mortality, and DALYs than females at all ages, with mortality/DALYs peaking in 55–64 age groups then declining (S1 Fig). OC's age/sex patterns were similar to TC's trends (S2 Fig).

## Risk factors

This study quantified temperature-attributable mortality and DALYs for TC, AC, and OC in 2021 globally and across 5 SDI/21 GBD regions. Except North Africa and the Middle East, low temperature contributed more to all three diseases: globally, it accounted for 11.8% (TC), 6.4% (AC), 5.3% (OC) of deaths and 10.8% (TC), 6.2% (AC), 4.6% (OC) of DALYs. In High SDI regions showed maximum contributions, low temperature caused 15.6% (TC), 7.8% (AC/OC) deaths and 14.3% (TC), 7.3% (AC), 7% (OC) DALYs, while Low SDI regions showed minimal contributions (deaths: 6.1% (TC), 4.2% (AC), 2.8% (OC); DALYs: 5.2% (TC), 2.9% (AC), 2.4% (OC)). Central Europe exhibited the highest low temperature impacts: 13.4% (TC), 9.5% (AC/OC) deaths, and 13.2% (TC), 9.3% (AC), 9.2%(OC) DALYs (Fig 7).

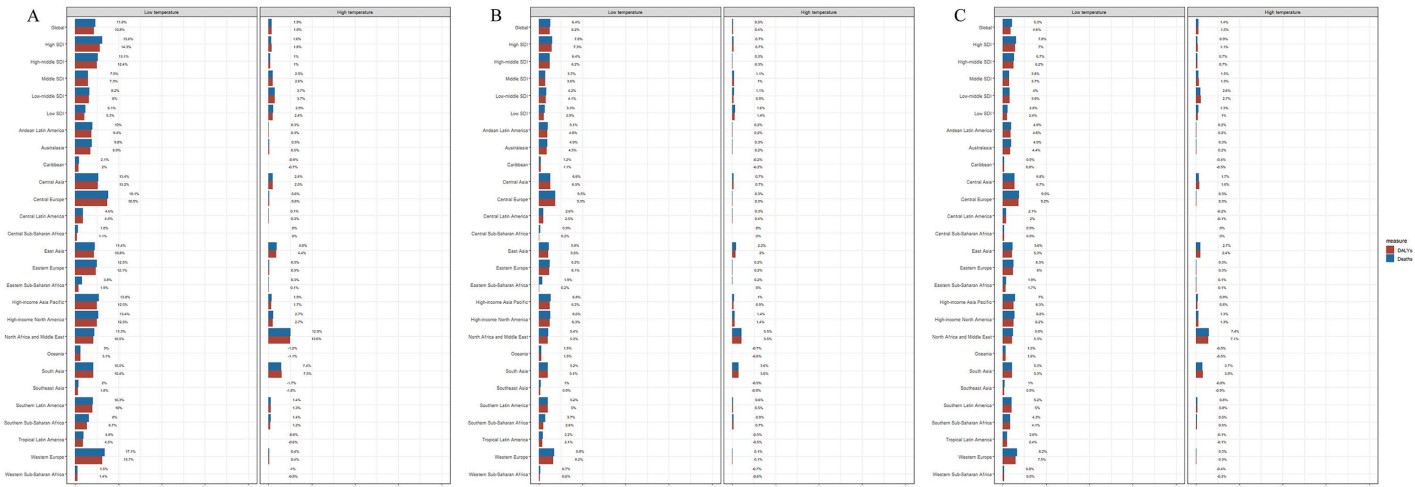

**Fig 7. Percentage of ASDRs and ASR_DALYs attributable to risk factors (low temperature and high temperature) in global and 21 GBD regions in 2021.** (A) Total cardiomyopathy. (B) Alcoholic cardiomyopathy. (C) Other cardiomyopathy.

## Forecasts for future disease trends of TC, AC, and OC

**TC Predictions.** Global ASPR will continue rising to 60.1/100,000 by 2040 (males: 73.8, females: 47.9), ASDR will decline to 3.4 (males: 4.8, females: 2.1), and ASR_DALYs will rise temporarily in 2022 before falling to 100.5 by 2040 (males: 146.9, females: 58.6), with males consistently higher across all metrics. What's more, the male ASPR growth rate exceeds that of females till 2040 (Fig 8).

**AC Predictions.** Global ASPR will decline to 3.6/100,000 by 2040 (males: 6.0, females: 1.4); ASDR temporarily rises in 2022 before dropping to 0.5 (males: 0.8, females: 0.2); ASR_DALYs follows the ASDR pattern, dropping to 16.1 by 2040 (males: 25.8, females: 6.8) (S9 Fig).

**OC Predictions.** Global ASPR will rise moderately (57.3/100,000 by 2040; males: 69.1, females: 47.1); ASDR maintains a decline to 3.0 (males: 4.1, females: 2.1); ASR_DALYs continue decreasing to 84.6 (males: 119.3, females: 53.5) (S10 Fig).

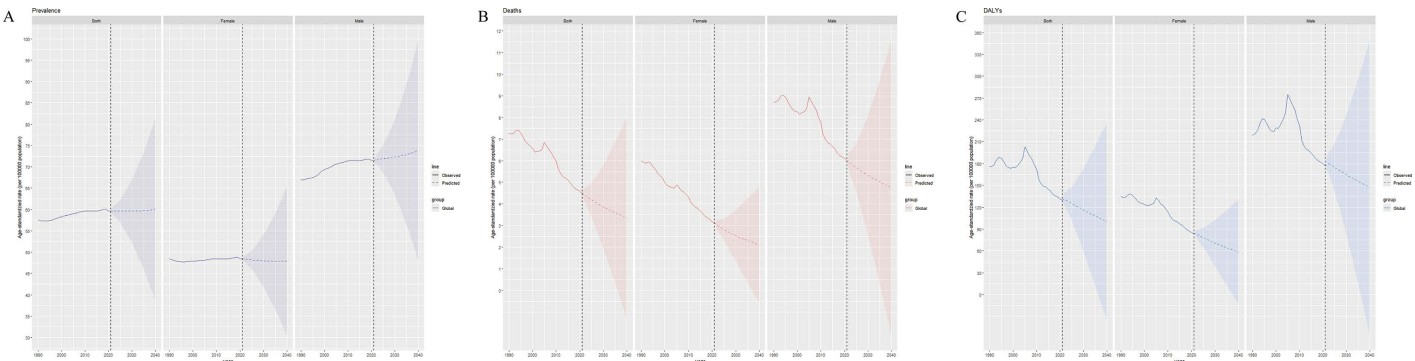

**Fig 8. Global disease trends predictions of total cardiomyopathy from 2021 to 2040.** (A) ASPRs. (B) ASDRs. (C) ASR_DALYs.

## Sensitive analysis

To assess the robustness of the Bayesian Age-Period-Cohort (BAPC) model's predictions, we conducted a sensitivity analysis using the autoregressive integrated moving average (ARIMA) time series model. For each key outcome indicator (prevalence, mortality, and DALYs), we fitted ARIMA models using the auto.arima function from the R language forecasting package (version 8.21.1) based on age-standardized rate data from 1990 to 2021. Model selection was based on minimizing the corrected Akaike Information Criterion (AICc), and predictions along with 95% prediction intervals for the period 2022–2040 were generated. The results of this sensitivity analysis are presented in the supplementary material (S11 Fig). The ARIMA model produces prediction results consistent with those of the BAPC model for the aforementioned indicators on TC, confirming the robustness of the BAPC model.

## Discussion

This study utilized GBD 2021 data to quantify the global burden of AC and OC, while leveraging AC/OC prevalence, mortality, DALYs profiles to derive TC-related epidemiological metrics (prevalence, mortality, DALYs) across 204 countries, 5 SDI strata, and 21 GBD regions. Stratified by year, age, gender, socioeconomic development (SDI), and geography, we computed age-standardized rates globally and regionally. From 1990 to 2021, TC, AC, and OC exhibited upward trends in prevalence, mortality, and DALYs, aligning with historical GBD findings [14]. Notably, TC/OC age-standardized prevalence rates (ASPR) increased globally, whereas OC's ASPR and TC/AC/OC's age-standardized mortality/DALYs (ASDR/ASR_DALYs) demonstrated divergent declines.

Regional analysis revealed convergent epidemiological patterns for TC and OC, with both diseases demonstrating their lowest age-standardized prevalence (ASPR), mortality (ASDR), and DALYs (ASR_DALYs) in Middle SDI regions. However, distinct divergences emerged: TC, AC, and OC exhibited their steepest declining trends in ASPR/ASDR/ASR_DALYs across High SDI and High-middle SDI strata, while paradoxically, TC/AC displayed peak ASPR in High SDI regions, whereas TC/AC's highest ASDR/ASR_DALYs shifted to High-middle SDI regions, and OC's ASDR/ASR_DALYs burdens peaked in Low SDI regions. Notably, three indicators for TC ranked second highest in Low SDI regions, similar to regional dominance for OC. Crucially, all three diseases followed inverted N-shaped trajectories, with ASPR/ASDR/ASR_DALYs first decreasing, then increasing as regional SDI indices rose to some extent, but decreasing lastly generally. These SDI disparities may be associated with intertwined healthcare resource allocation, socioeconomic stratification, risk factor exposures, and data reliability. High SDI regions, despite abundant healthcare resources, may face modern risk exposures (e.g., sedentary lifestyles, pollution) that paradoxically correlate with elevated ASPR through overdiagnosis and delayed mortality, suggesting potential needs for targeted chronic disease management and cardiomyopathy prevention in high-risk demographics. Conversely, High SDI regions appear to achieve EAPC stabilization/decrease through integrated healthcare optimization, risk mitigation strategies, and robust public health frameworks, indicating possible effective burden mitigation. High-middle SDI regions may grapple with resource maldistribution, creating a "high burden-low efficiency" paradox where fragmented healthcare access is associated with amplified ASDR/DALY burdens. Low SDI regions appear to face a vicious cycle where inadequate health literacy and preventive care are correlated with delayed diagnosis until advanced disease stages, potentially exacerbating the ASPR/ASDR/DALY triple burdens. Middle SDI regions, undergoing epidemiological transition, exhibit intermediate burden profiles shaped by partial healthcare improvements and residual risk exposures. These findings highlight potential considerations for proactive policy interventions, including primary healthcare expansion and health education campaigns, to potentially optimize resource allocation and address these patterns, thereby potentially curbing future disease burdens. This SDI-stratified analysis underscores the need for tailored strategies addressing unique regional epidemiological profiles [30,31].

Regional analysis of 21 GBD regions revealed distinct cardiomyopathy burden patterns: Sub-Saharan Africa exhibited elevated OC prevalence (ASPR), mortality (ASDR), and DALYs (ASR_DALYs), which may be attributable to fragmented primary/secondary prevention systems and healthcare inefficiencies amid lifestyle-driven epidemiological transitions [32].

Conversely, Eastern Europe showed peak TC/AC burdens, corroborating national-level findings where Russia, Ukraine, and Latvia reported extreme AC maxima for all three indicators. WHO Regional Office data (2019) revealed Europe's highest alcohol consumption per capita (15+years) at 15.2L pure alcohol, with Eastern Europe demonstrating a declining prevalence trend stabilized within ±3% over two decades. Disparities persisted between European income tiers: low/middle-income regions exhibited 4.8L higher consumption than high-income counterparts, aligning with our observed regional alcohol burden patterns [33]. Despite the WHO's 2030 targets to curb harmful alcohol use, European policy implementation appears inadequate—alcohol warning labels are scarce outside Western Europe, and public awareness campaigns are lacking, suggesting challenges in harm reduction strategies [34–36]. In contrast, East Asia demonstrated the lowest TC/OC burdens, reflecting regional heterogeneity in socioeconomic development and health literacy [37]. This underscores the need for region-specific interventions targeting cardiomyopathy risk trajectories. Notably, East Asian governments have implemented multifaceted policies: Japan and China integrate cardiovascular-risk food labeling, taxation on tobacco/alcohol, and population-wide health promotion initiatives emphasizing physical activity and dietary modifications [38]. These proactive measures contrast sharply with Eastern Europe's policy inertia, highlighting a correlation between structured public health frameworks and potentially mitigated cardiomyopathy burdens.

This study revealed disproportionately elevated ASPR, ASDR, and ASR_DALYs for TC/OC in the <5 age cohort compared to adolescent groups, with infantile cardiomyopathy exhibiting distinct epidemiological divergence—1-year-olds showed 3.2-fold higher incidence rates than older children (1–18 years), which may reflect genetic predisposition prevalence [39]. Post-5 age groups demonstrated age-related ascension of TC/OC indicators, potentially amplified by shifting disease spectra amid global population aging [40]. Gender disparities were most pronounced for AC, where males demonstrated elevated prevalence rates, aligning with male-predominant alcohol consumption patterns documented in the 2019 AC GBD study [41]. Although the ratio of male to female per capita alcohol consumption varies due to cultural and economic factors, WHO global alcohol consumption data revealed a persistent male dominance in drinking patterns, with the ratio typically ranging from 1.2:1 to nearly 2:1 in 2019 [33]. This male-dominant pattern suggests a potential association between behavioral risks and disease manifestation, particularly in high-risk cardiomyopathy phenotypes.

In risk factor studies, the results showed that ASPR, ASDR, and ASR_DALYs for cardiomyopathy were attributable to both low and high temperature indices. Existing literature includes many related studies on the risk factors of low temperature and high temperature. Current evidence suggests that high or low ambient temperature can cause abnormal body temperature, which may be associated with the deterioration of existing cardiomyopathy or stress cardiomyopathy [20,42,43]. Ambient temperature fluctuations may modulate alcohol-related behavioral risk exposure frequencies, potentially influencing AC disease trends. Current GBD data remains inconclusive on this ambient temperature-alcohol consumption relationship, necessitating deeper investigation.

We are familiar with the global pandemic of COVID-19 that occurred from 2019 to 2021, which may have affected the analysis of the global trends in cardiomyopathy. Additionally, many studies have discussed the changes in the prevalence of cardiomyopathy caused by COVID-19 and its pandemic. As shown in this study, high burdens of cardiomyopathy continue in Eastern Europe and other regions, consistent with reports of severe impacts of COVID-19 in these areas [44]. Some studies suggest that complications of COVID-19 can manifest in various cardiovascular ways, including myocardial injury, arrhythmias, cardiac arrest, heart failure, and coagulation abnormalities [44,45]. Furthermore, the infection rate, hospitalization rate, and mortality rate of SARS-CoV-2 in diabetic populations are higher than in the general population, making subgroups with cardiomyopathy in diabetic patients potentially more vulnerable to severe prognosis difficulties and worse disease outcomes after contracting COVID-19 [46]. These studies highlight the potential impact of COVID-19 on the prevalence and severity of cardiomyopathy during the pandemic, emphasizing the importance of considering healthcare facilities and related health policies after the pandemic.

In this study, we calculated the disease burden of TC using data from AC and OC in the 2021 GBD database. This analysis has clear limitations. First, our data sources all rely on GBD 2021, although the GBD study strives for the

greatest reliability and comparability in the data included, it is inevitably subject to reporting delays or inaccuracies, inconsistent data collection or source quality, classification errors, and country-specific coding biases. For example, in low-income countries, cardiomyopathy may not receiv sufficient attention, possibly leading to underestimation of the disease burden in these regions. Additionally, the classification of OC is very vague, the case definitions of cardiomyopathy and its subtypes may vary across different sources and study time spans, which may introduce heterogeneity, and the GBD database does not further categorize cardiomyopathy types, making it impossible to conduct targeted analysis of related disease indicators based on specific types of cardiomyopathy. Furthermore, when using GBD databases from different years for analysis, changes in data sources and processing methods may lead to partial differences in our results compared to past GBD studies on cardiomyopathy. Moreover, the GBD database does not query incidence data for AC and OC, which is also a significant reason for the incompleteness of our study content. What's more, our predictive approach, although based on robust statistical models, cannot account for unforeseen changes in future risk factors, diagnostic techniques, or treatment methods that may significantly alter the disease trend. Finally, the COVID-19 pandemic's impact on healthcare institutions may have affected GBD data collection for cardiomyopathy-related indicators, which could be a factor in the increased burden of cardiomyopathy. However, since GBD classifies COVID-19 as a separate disease, we cannot explore the specific impact of this risk factor on changes in cardiomyopathy indicators.

## Conclusion

In conclusion, the disease burden of cardiomyopathy in the world is still not optimistic, and there are many kinds of cardiomyopathy, and its epidemic characteristics in different regions of the world also have significant differences, so corresponding disease control measures need to be taken according to different geographical differences, cultural differences and economic differences. Similarly, the prevalence of cardiomyopathy has been effectively controlled in some regions, and these successful experiences should be used to formulate health policies. For alcohol use, countries should introduce relevant control policies, especially in Eastern European countries such as Russia. In future analysis of cardiomyopathy, it is needed to further classify OC from congenital or acquired cardiomyopathy to achieve not only treatment or better prognosis, but also prevention of more occurrence of cardiomyopathy at the genes, environment, and lifestyle levels level, and more research on the disease burden of cardiomyopathy attributed to other risk factors, to effectively control the future burden of cardiomyopathy.

## Supporting information

**S1 Fig. The age-specific numbers and ASRs of alcoholic cardiomyopathy by SDI regions in 2021.** (A) ASPRs. (B) ASDRs. (C) ASR_DALYs.
(DOCX)

**S2 Fig. The age-specific numbers and ASRs of other cardiomyopathy by SDI regions in 2021.** (A) ASPRs. (B) ASDRs. (C) ASR_DALYs.
(DOCX)

**S3 Fig. Socio-demographic index trends with ASRs of alcoholic cardiomyopathy in 21 GBD regions in 2021.** (A) ASPRs. (B) ASDRs. (C) ASR_DALYs.
(DOCX)

**S4 Fig. Socio-demographic index trends with ASRs of other cardiomyopathy in 21 GBD regions in 2021.** (A) ASPRs. (B) ASDRs. (C) ASR_DALYs.
(DOCX)

**S5 Fig. ASRs of alcoholic cardiomyopathy in 204 countries and territories in 2021.** (A) ASPRs. (B) ASDRs. (C) ASR_DALYs.
(DOCX)

**S6 Fig. ASRs of other cardiomyopathy in 204 countries and territories in 2021.** (A) ASPRs. (B) ASDRs. (C) ASR_DALYs.
(DOCX)

**S7 Fig. Socio-demographic index trends with ASRs of alcoholic cardiomyopathy in 204 countries and territories in 2021.** (A) ASPRs. (B) ASDRs. (C) ASR_DALYs.
(DOCX)

**S8 Fig. Socio-demographic index trends with ASRs of other cardiomyopathy in 204 countries and territories in 2021.** (A) ASPRs. (B) ASDRs. (C) ASR_DALYs.
(DOCX)

**S9 Fig. Global disease trends predictions of alcoholic cardiomyopathy from 2021 to 2040.** (A) ASPRs. (B) ASDRs. (C) ASR_DALYs.
(DOCX)

**S10 Fig. Global disease trends predictions of other cardiomyopathy from 2021 to 2040.** (A) ASPRs. (B) ASDRs. (C) ASR_DALYs.
(DOCX)

**S11 Fig. Global disease trends predictions of total cardiomyopathy from 2021 to 2040 using ARIMA.**
(DOCX)

**S1 Table. 1990–2021 Global and regional prevalence trends in alcoholic cardiomyopathy burden.**
(DOCX)

**S2 Table. 1990–2021 Global and regional mortality trends in alcoholic cardiomyopathy burden.**
(DOCX)

**S3 Table. 1990–2021 Global and regional DALYs trends in alcoholic cardiomyopathy burden.**
(DOCX)

**S4 Table. 1990–2021 Global and regional prevalence trends in other cardiomyopathy burden.**
(DOCX)

**S5 Table. 1990–2021 Global and regional mortality trends in other cardiomyopathy burden.**
(DOCX)

**S6 Table. 1990–2021 Global and regional DALYs trends in other cardiomyopathy burden.**
(DOCX)

**S7 Table. 1990–2021 Global and national prevalence trends in total cardiomyopathy burden.**
(DOCX)

**S8 Table. 1990–2021 Global and national mortality trends in total cardiomyopathy burden.**
(DOCX)

**S9 Table. 1990–2021 Global and national DALYs trends in total cardiomyopathy burden.**
(DOCX)

**S10 Table. 1990–2021 Global and national prevalence trends in alcoholic cardiomyopathy burden.**
(DOCX)

**S11 Table. 1990–2021 Global and national mortality trends in alcoholic cardiomyopathy burden.**
(DOCX)

**S12 Table. 1990–2021 Global and national DALYs trends in alcoholic cardiomyopathy burden.**
(DOCX)

**S13 Table. 1990–2021 Global and national prevalence trends in other cardiomyopathy burden.**
(DOCX)

**S14 Table. 1990–2021 Global and national mortality trends in other cardiomyopathy burden.**
(DOCX)

**S15 Table. 1990–2021 Global and national DALYs trends in other cardiomyopathy burden.**
(DOCX)

**S1 Data. The dataset used in the article.**
(ZIP)

## Acknowledgments

Thanks to the GBD 2021 database for data provision.

## Author contributions

**Conceptualization:** Haoyang Chen.

**Data curation:** Haoyang Chen.

**Formal analysis:** Haoyang Chen.

**Funding acquisition:** Yanzhang Tian.

**Investigation:** Haoyang Chen.

**Methodology:** Haoyang Chen.

**Project administration:** Ruifeng Liang, Yanzhang Tian.

**Resources:** Haoyang Chen.

**Software:** Haoyang Chen.

**Supervision:** Ruifeng Liang, Yanzhang Tian.

**Validation:** Haoyang Chen, Ruifeng Liang, Yanzhang Tian.

**Visualization:** Haoyang Chen.

**Writing – original draft:** Haoyang Chen.

**Writing – review & editing:** Haoyang Chen, Ruifeng Liang, Yanzhang Tian.

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
