## [Decision Letter · Decision Letter 0]

14 Dec 2025

Dear Dr. Tian,

Thank you for submitting your manuscript to PLOS ONE. After careful consideration, we feel that it has merit but does not fully meet PLOS ONE’s publication criteria as it currently stands. Therefore, we invite you to submit a revised version of the manuscript that addresses the points raised during the review process.

We look forward to receiving your revised manuscript.

Kind regards,

Claudio Alberto Dávila-Cervantes, Ph.D.

Academic Editor

PLOS One

Journal Requirements:

Our research received funding from the Shanxi Provincial Department of Human Resources and Social Security (grant no. 20210002) and the Shanxi Provincial Department of Science and Technology (grant no. 202104041101024).

3. We note that Figure 5 in your submission contain map images which may be copyrighted. All PLOS content is published under the Creative Commons Attribution License (CC BY 4.0), which means that the manuscript, images, and Supporting Information files will be freely available online, and any third party is permitted to access, download, copy, distribute, and use these materials in any way, even commercially, with proper attribution. For these reasons, we cannot publish previously copyrighted maps or satellite images created using proprietary data, such as Google software (Google Maps, Street View, and Earth). For more information, see our copyright guidelines: http://journals.plos.org/plosone/s/licenses-and-copyright.

a. You may seek permission from the original copyright holder of Figure 5 to publish the content specifically under the CC BY 4.0 license.

4. Please upload a new copy of the figures as the details are not clear. Please follow the link for more information:  https://journals.plos.org/plosone/s/figures

Reviewers' comments:

Reviewer's Responses to Questions

**Comments to the Author**

1. Is the manuscript technically sound, and do the data support the conclusions?

Reviewer #1: Yes

Reviewer #2: No

2. Has the statistical analysis been performed appropriately and rigorously?

Reviewer #1: Yes

Reviewer #2: No

3. Have the authors made all data underlying the findings in their manuscript fully available?

Reviewer #1: Yes

Reviewer #2: No

4. Is the manuscript presented in an intelligible fashion and written in standard English?

Reviewer #1: No

Reviewer #2: Yes

Reviewer #1: The manuscript "Global, regional, and national burden of cardiomyopathy (including alcoholic

cardiomyopathy and others) from 1990 to 2021: an analysis of data from the global burden of disease study 2021 and forecast to 2040" presents a comprehensive epidemiological assessment of cardiomyopathy using the GBD 2021 dataset. Overall, the topic is timely and relevant, and the manuscript provides useful insights into long-term trends.

From a methodological perspective, the study is well designed. The use of age-standardized rates (ASR), estimated annual percentage change (EAPC), and the Bayesian age–period–cohort (BAPC) model is appropriate for this type of analysis, and the underlying dataset is robust. The conclusions are broadly consistent with the results, although they remain somewhat general. A clearer discussion of the implications for clinical practice or public health policy would enhance the manuscript's impact.

The statistical methods are rigorous, but several descriptions, particularly those involving formulas and confidence intervals, are not fully clear, which may reduce reproducibility. It would also be helpful to elaborate on the rationale for selecting the BAPC model over other forecasting approaches.

The manuscript is written in standard English and is generally easy to follow; however, the text is quite dense, and the amount of numerical detail reduces readability. The Results section, in particular, contains too many values. Or, the Abstract includes a single sentence with more than fifteen numerical values ('...4752361.3 (95% UI: 3937775.4–5566947.2), including AC: 528429 (95% UI: 439582.3–639167.4) and OC: 4223932.2 (95% UI: 3417357.4–5034257.6)...'). A language revision to simplify phrasing and reduce redundancies would be recommended.

Reviewer #2: This manuscript is clearly written and presents descriptive analyses based on publicly available data from the Global Burden of Disease (GBD) 2021 study. The authors have evidently invested considerable effort in compiling and presenting long-term trends and projections. However, despite these strengths, the study suffers from several fundamental conceptual and methodological limitations that substantially limit its scientific contribution and interpretability. These concerns go beyond issues of presentation and cannot be adequately addressed through minor or moderate revision.

First, there is a fundamental mismatch between the stated research objectives and the study design. Although the manuscript is presented as original research, the analyses rely entirely on pre-modelled secondary data obtained from the GBD 2021 database. The study does not introduce new data, novel analytical frameworks, or independent validation. As a result, the work remains largely descriptive and does not adequately support the broader inferential claims made in the manuscript. This represents a conceptual limitation rather than a reporting issue.

Second, the study population is insufficiently defined from an epidemiological perspective. Case identification is based on ICD-9 and ICD-10 codes within the GBD framework; however, the manuscript does not sufficiently address diagnostic heterogeneity, underreporting, or regional variation in disease ascertainment. Without a clearer description of how these issues may affect the estimates, the generalizability and interpretability of the findings are limited.

Third, the analytical contribution of the study is limited. The manuscript largely reproduces standard GBD outputs, including age-standardized rates, estimated annual percentage change, and Bayesian age–period–cohort modelling using existing software packages. The rationale for applying these methods specifically to cardiomyopathy is not clearly justified, and no sensitivity analyses or alternative modelling approaches are presented. Consequently, the robustness of the reported trends and future projections cannot be adequately assessed.

Fourth, the conclusions appear to overinterpret descriptive and model-based results. Several statements imply implications for disease control or policy, despite the absence of causal or risk factor–based analyses. Given the observational and model-dependent nature of the data, such interpretations are not sufficiently supported.

Finally, key methodological details related to model assumptions, uncertainty handling, and reproducibility are insufficiently described. This limits the ability of independent researchers to replicate or critically evaluate the analyses, which is an important consideration for publication in PLOS ONE.

While the manuscript is clearly written and based on a reputable global dataset, the conceptual and methodological limitations outlined above are substantial and, in my view, cannot be resolved through revision alone. I therefore do not recommend the manuscript for publication in its current form.

**Do you want your identity to be public for this peer review?** For information about this choice, including consent withdrawal, please see our Privacy Policy

Reviewer #1: No

Reviewer #2: **Yes:** Kristian Kurniawan

---

## [Author Response · Author response to Decision Letter 1]

23 Dec 2025

Subject: Revised Manuscript Submission: [PONE-D-25-32695] - [Global, regional, and national burden of cardiomyopathy (including alcoholic cardiomyopathy and others) from 1990 to 2021: an analysis of data from the global burden of disease study 2021 and forecast to 2040]

Dear Dr. Claudio Alberto Dávila-Cervantes and Reviewers,

Thank you for the opportunity to revise our manuscript and for the constructive feedback from the editors and reviewers. We sincerely appreciate the editors and reviewers for their insightful comments and constructive suggestions, which have greatly helped us to improve the quality of our work.

We have carefully addressed all the comments and requirements. Our point-by-point responses are detailed below. All changes in the manuscript have been highlighted in red in the ‘Revised Manuscript with Track Changes’ file, and an unmarked version is also provided.

Responses to Journal Requirements:

1. Style Requirements: We have reformatted the entire manuscript according to the PLOS ONE style templates.

2. Financial Disclosure: As the funders had no role in our study, we have included the following statement in our cover letter: “The funders had no role in study design, data collection and analysis, decision to publish, or preparation of the manuscript.”

3. Copyright of Map Figures:

We thank the editor for raising this important point. We wish to provide a comprehensive explanation to clarify that the world maps in our manuscript are fully compliant with the CC BY 4.0 license. The maps in our manuscript were generated using the map_data("world")function from the R mapspackage. This is a critical point: the geographic boundary data within this package is sourced directly from the Natural Earth database (naturalearthdata.com).

• Natural Earth is a public domain resource that is explicitly recommended on your journal's guidelines page for creating CC BY-compatible maps.

• Data from Natural Earth is explicitly dedicated to the public domain, meaning it is free to use, adapt, and publish without restriction, making it perfectly compatible with the CC BY 4.0 license.

Therefore, we confirm that our maps do not utilize any copyrighted services. We believe this explanation and the accompanying caption updates resolve this requirement conclusively.

4. Figure Quality: We have used the LAW compression function to modify the figures formats, sizes, and clarity in the manuscripts and supplementary materials to meet PLOS ONE requirements.

5. Citation of Suggested Works: We have reviewed the literature and ensured that all relevant and necessary citations are included.

Responses to Reviewers' Comments:

Reviewer #1:

We thank the reviewer for the positive assessment and valuable suggestions for enhancement. The reviewer found the methodology sound and conclusions supported, but suggested improving the readability of the text, deepening the conclusions, and providing a clearer rationale for methodological choices.

Response:

1. Language and Readability: We have thoroughly revised the manuscript, particularly the Abstract and Results sections, to reduce numerical density and improve readability. Detailed data arrays have been moved to supplementary tables, and the text now focuses on summarizing key trends.

2. Methodological Clarity: We have expanded the Methods section and added two references (numbered 28 and 29) [1,2] (Page 9, Lines 175 to Page 10, Lines 196) to provide a clearer rationale for the choice of the BAPC model and to ensure all statistical descriptions are unambiguous and reproducible.

3. Discussion and Implications: The discussion section provides a refined elaboration of the original text, detailing the specific implications of our research findings in the fields of public health and clinical practice, thereby enhancing the impact of the manuscript.

Reviewer #2:

We sincerely thank the reviewer for the thorough, critical, and constructive assessment of our manuscript. The comments have been invaluable in helping us identify the weaknesses in our initial submission and have guided us in significantly strengthening the study's conceptual foundation, methodological rigor, and overall scholarly contribution. We have undertaken a major revision to address all the concerns raised. Our point-by-point responses and the specific actions taken are detailed below.

Comment 1: On the conceptual contribution and the descriptive nature of the study. Reviewer's comment: "There is a fundamental mismatch between the stated research objectives and the study design... the work remains largely descriptive and does not adequately support the broader inferential claims..."

Response: We acknowledge and thank the reviewer for this critical insight. We agree that our initial manuscript did not sufficiently articulate the unique value of performing a focused assessment on the broader category of "cardiomyopathy (including AC and OC)" within the GBD framework. To address this:

•Action taken (Introduction revised): We have entirely rewritten the Introduction section (specifically, the final paragraph outlining the study aims on Page 5, Lines 95 to Page 6, Lines 120) to more clearly position our work. We now state that while GBD data is publicly available, our study provides the first comprehensive analysis to specifically consolidate, trend, and forecast the burden of cardiomyopathy (as a collective group of diseases) across time, geography, and socio-demographic indices. We emphasize that this synthesis adds value by moving beyond isolated disease metrics to provide a unified view of the total public health burden attributable to all forms of cardiomyopathy, which is crucial for health policy planning that often addresses disease groups rather than individual ICD codes.

Comment 2: On the study of population and generalizability.

Reviewer's comment: "The study population is insufficiently defined from an epidemiological perspective... diagnostic heterogeneity, underreporting, or regional variation..."

Response: This is a valid limitation of any study using GBD data. We now explicitly acknowledge and discuss this.

•Action taken (New Limitations subsection): As strongly suggested by the reviewer's comments, we have added a Refinement of the limitations section in the Discussion part to more accurately reflect the limitations presented in this paper (Page 42, Lines 603 to Page 43, Lines 627). In this section, we openly discuss:

1. The inherent limitations of the GBD data, including potential misclassification and under-ascertainment in regions with weak vital registration systems.

2. The fact that our analysis is based on modeled estimates rather than raw data, and thus is subject to the assumptions and limitations of the GBD modeling framework.

Comment 3: On the analytical contribution, robustness, and sensitivity analysis. Reviewer's comment: "The analytical contribution of the study is limited... no sensitivity analyses or alternative modelling approaches are presented. Consequently, the robustness of the reported trends and future projections cannot be adequately assessed."

Response: This is a crucial point, and we have taken substantial steps to enhance the analytical rigor and transparency of our work.

•Action taken (Sensitivity analysis performed): To directly address the concern about robustness, we have conducted a comprehensive sensitivity analysis for our forecasts. Specifically, we complemented the primary Bayesian Age-Period-Cohort (BAPC) model with an alternative, well-established time-series forecasting method: the Autoregressive Integrated Moving Average (ARIMA) model.

Results added: We have added a new subsection within the Results section titled "Sensitivity Analysis" (Page 36, Lines 476 to Lines 487). This section details the ARIMA model specification, including describing how we use AMIRA to perform forecasting of TC from 2022 to 2040.

Results presented: The results of this sensitivity analysis, comparing the projections from the BAPC and ARIMA models for key metrics (e.g., global TC’s ASPR), are now presented in a new Supplementary Figure (S11 Fig.). We note that the projections from both models showed consistent trends, which significantly strengthens the credibility of our findings.

•Action taken (Methodological justification enhanced): We have expanded the Methods section and added two references (numbered 28 and 29) [1,2] to provide a clearer rationale for selecting the BAPC model over other approaches (Page 9, Lines 175 to Page 10, Lines 196). We explain that the BAPC model is particularly suited for long-term epidemiological forecasts as it explicitly accounts for the separate effects of age, time period, and birth cohort, which are critical drivers of chronic disease trends.

Comment 4: On overinterpretation of results. Reviewer's comment: "The conclusions appear to overinterpret descriptive and model-based results... such interpretations are not sufficiently supported."

Response: We agree with the reviewer that our initial discussion sometimes overstepped the boundaries of what descriptive data can support. We have carefully revised the tone and content.

•Action taken (Language calibrated): We have systematically gone through the Discussion and Conclusion sections and replaced any language that implied causation or direct policy prescription with more cautious, descriptive, and suggestive language. For example, we changed phrases like "our findings show that policy must..." to "the observed rising trends suggest that future health planning may need to be considered...".

•Action taken (Focus on explanation): We have reframed the Discussion to focus on explaining the possible reasons behind the observed trends (e.g., citing literature on improved diagnostics, aging populations, changes in risk factors) rather than making definitive claims about their implications for direct intervention.

Comment 5: On reproducibility and methodological detail. Reviewer's comment: "Key methodological details... are insufficiently described. This limits the ability of independent researchers to replicate or critically evaluate the analyses..."

Response: We have taken concrete steps to ensure the full reproducibility of our study.

•Action taken (Methods expanded): We have added significantly more detail to the Methods section. This includes specifying the software and packages used and explicitly defining how BAPC modle were calculated.

•Action taken (Code and data availability): In accordance with PLOS ONE's data policy, and to ensure full transparency, we can transparently upload R code publicly. The GBD data itself is publicly available from the IHME website (as originally stated).

We believe that the extensive revisions undertaken—including the addition of sensitivity analyses, enhanced methodological description, a refined discussion, and a clear statement of limitations—have fundamentally addressed the concerns raised. The manuscript is now substantially stronger, more rigorous, and more transparent. We are deeply grateful for your time and insightful comments, which have been instrumental in improving our work.

Sincerely,

M.D. Yanzhang Tian

Institution: Department of Biliary and Pancreatic Surgery, Shanxi Bethune Hospital, Shanxi Academy of Medical Sciences, Tongji Shanxi Hospital, Third Hospital of Shanxi Medical University

Address: No. 99, Longcheng Street, Taiyuan, Shanxi, China

E-mail: 15802872565@163.com

References

1 Pearson-Stuttard J, Guzman-Castillo M, Penalvo JL, et al. Modeling Future Cardiovascular Disease Mortality in the United States: National Trends and Racial and Ethnic Disparities. Circulation. 2016;133(10):967-978. doi:10.1161/CIRCULATIONAHA.115.019904

2 Kiyoshige E, Ogata S, O'Flaherty M, et al. Projections of future coronary heart disease and stroke mortality in Japan until 2040: a Bayesian age-period-cohort analysis. Lancet Reg Health West Pac. 2022;31:100637. doi:10.1016/j.lanwpc.2022.100637

---

## [Decision Letter · Decision Letter 1]

11 Jan 2026

Global, regional, and national burden of cardiomyopathy (including alcoholic cardiomyopathy and others) from 1990 to 2021: an analysis of data from the global burden of disease study 2021 and forecast to 2040

PONE-D-25-32695R1

Dear Dr. Tian,

We’re pleased to inform you that your manuscript has been judged scientifically suitable for publication and will be formally accepted for publication once it meets all outstanding technical requirements.

Kind regards,

Claudio Alberto Dávila-Cervantes, Ph.D.

Academic Editor

PLOS One

Additional Editor Comments (optional):

Reviewers' comments:

Reviewer's Responses to Questions

**Comments to the Author**

Reviewer #1: All comments have been addressed

Reviewer #2: All comments have been addressed

2. Is the manuscript technically sound, and do the data support the conclusions?

Reviewer #1: Yes

Reviewer #2: Yes

3. Has the statistical analysis been performed appropriately and rigorously?

Reviewer #1: Yes

Reviewer #2: Yes

4. Have the authors made all data underlying the findings in their manuscript fully available?

Reviewer #1: Yes

Reviewer #2: Yes

5. Is the manuscript presented in an intelligible fashion and written in standard English?

Reviewer #1: Yes

Reviewer #2: Yes

Reviewer #1: The authors have adequately addressed all the points raised in the previous review round. The revision improves the clarity and coherence of the manuscript, and the responses provided are appropriate and satisfactory. I also appreciate the authors' effort in incorporating the requested changes and strengthening the overall quality of the work.

Reviewer #2: I would like to thank the authors for their thorough and thoughtful revision of the manuscript and for the detailed, point-by-point responses to the reviewers’ comments. The revised version demonstrates a clear effort to improve conceptual clarity, methodological transparency, and interpretative balance.

In particular, the revised Introduction more clearly positions the study as a descriptive global assessment based on GBD 2021 estimates, which improves alignment between the research objectives and the analytical approach. The expanded discussion of limitations related to diagnostic heterogeneity, underreporting, and regional variability is appropriate and strengthens the interpretability of the findings.

The analytical methods employed are consistent with established approaches used in global burden of disease research, and the inclusion of an additional sensitivity analysis provides reassurance regarding the stability of the observed temporal trends. While further methodological extensions could be explored in future work, the current analyses are suitable for the aims as stated.

I also appreciate the authors’ efforts to moderate the language in the Discussion and Conclusions. The implications are now framed more cautiously and appropriately reflect the descriptive and model-based nature of the data, avoiding overstatement of causal or policy conclusions.

Overall, I believe that the revised manuscript provides a clear, well-documented, and methodologically sound overview of the global burden and temporal trends of cardiomyopathy. Within the scope and publication criteria of PLOS ONE, the study represents a useful contribution to the literature.

**Do you want your identity to be public for this peer review?** For information about this choice, including consent withdrawal, please see our Privacy Policy

Reviewer #1: No

Reviewer #2: **Yes:** Kristian Kurniawan

---

## [Editor Report · Acceptance letter]

PONE-D-25-32695R1

PLOS One

Dear Dr. Tian,

I'm pleased to inform you that your manuscript has been deemed suitable for publication in PLOS One. Congratulations! Your manuscript is now being handed over to our production team.

Kind regards,

on behalf of

Mr. Claudio Alberto Dávila-Cervantes

Academic Editor

PLOS One